# Pharmacological induction of selective endoplasmic reticulum retention as a strategy for cancer therapy

Mohamed Mahameed[1], Shatha Boukeileh[1], Akram Obiedat[1], Odai Darawshi[1], Priya Dipta[1], Amit Rimon[1], Gordon McLennan [2], Rosi Fassler[3], Dana Reichmann [3], Rotem Karni [4], Christian Preisinger[5], Thomas Wilhelm[6], Michael Huber [6] & Boaz Tirosh [1]✉

The integrated stress response (ISR) converges on eIF2α phosphorylation to regulate protein synthesis. ISR is activated by several stress conditions, including endoplasmic reticulum (ER) stress, executed by protein kinase R-like endoplasmic reticulum kinase (PERK). We report that ER stress combined with ISR inhibition causes an impaired maturation of several tyrosine kinase receptors (RTKs), consistent with a partial block of their trafficking from the ER to the Golgi. Other proteins mature or are secreted normally, indicating selective retention in the ER (sERr). sERr is relieved upon protein synthesis attenuation and is accompanied by the generation of large mixed disulfide bonded complexes, including ERp44. sERr was pharmacologically recapitulated by combining the HIV-protease inhibitor nelfinavir with ISRIB, an experimental drug that inhibits ISR. Nelfinavir/ISRIB combination is highly effective to inhibit the growth of RTK-addicted cell lines and hepatocellular (HCC) cells in vitro and in vivo. Thus, pharmacological sERr can be utilized as a modality for cancer treatment.

[1] Institute for Drug Research, The Hebrew University of Jerusalem, Jerusalem, Israel. [2] Lerner Research Institute, Cleveland Clinic Foundation, Cleveland, OH, USA. [3] The Alexander Silberman Institute of Life Sciences, The Hebrew University of Jerusalem, Safra Campus Givat Ram, Jerusalem, Israel. [4] Department of Biochemistry, Faculty of Medicine, IMRIC, The Hebrew University of Jerusalem, Jerusalem, Israel. [5] Proteomics Facility, IZKF Aachen, RWTH Aachen University, Aachen, Germany. [6] Institute of Biochemistry and Molecular Immunology, Medical School, RWTH Aachen University, Aachen, Germany. ✉email: boazt@ekmd.huji.ac.il

The unfolded protein response (UPR) is activated in response to perturbations in endoplasmic reticulum (ER) homeostasis. In mammalian cells, the UPR is comprised of at least three signaling pathways initiated by ER resident sensors: IRE1, PERK, and ATF6. IRE1 and PERK have self-kinase activities that regulate effector functions. IRE1 is an endoribonuclease (RNase) and a Ser/Thr kinase, while PERK is an eIF2α kinase. The phosphorylation of eIF2α attenuates global protein translation by sequestering the multisubunit GEF eIF2B, which is needed to charge the preinitiation complex with GTP. This phosphorylation leads to preferred translation of selective mRNAs, such as the one encoding the transcription factor ATF4[1] that activates a transcription program controlling cell survival and cellular metabolism.

PERK is one of four eIF2α kinases. The other three, PKR, GCN2, and HRI, are activated by different stress conditions, such as viral infection, lack of amino acids, and iron depletion, respectively. Thus, phosphorylation of eIF2α funnels multiple stress pathways to which the term "integrated stress response (ISR)" was coined[2].

Cellular and animal models using gain and loss of function of various UPR proteins have shown potential involvement of the UPR in major pathologies, such as diabetes, neurodegeneration, and cancer[3–5]. This has promoted the development of drugs that probe different elements of UPR signaling, hoping to identify potential disease modulators. Recognizing the importance of the PERK pathway in cancer, high-affinity inhibitors have been developed by several pharmaceutical companies. The first, GSK2606414 (termed here GSK414 in short), showed toxicity to the pancreas and loss of weight in preclinical models, which led to cessation of its further development for clinical applications[6]. Additional PERK inhibitors have been developed, all still in preclinical stages. A second inhibitor of the PERK pathway is ISRIB, which does not interact with PERK or any of the other three enzymes that phosphorylate eIF2α. Rather, ISRIB reverts the translation inhibition downstream of eIF2α phosphorylation by enhancing the GEF activity of eIF2B. This enhances the levels of the preinitiation translation complex. Preclinical analyses suggest that ISRIB is less toxic than GSK414 and may be suitable for human use[7]. ISRIB and additional analogs thereof have been developed for memory loss associated with brain trauma, neurodegeneration or white matter loss disease[8].

Half of melanoma tumors harbor the V600E mutation in BRAF, which sensitizes these tumors to specific BRAF inhibitors. PERK mutants identified in human melanoma are hypomorphic with dominant inhibitory function. A personalized approach for PERK inhibitors in cancer was proposed in light of strong pharmacological evidence that PERK inhibitors as single agents have profound anticancer efficacy against the BRAFV600E-dependent tumors[9]. Prompted by this discovery we analyzed BRAF mutated Mel526 cells for total phospho-tyrosine (P-Tyr) levels under conditions of ER stress in the presence and absence of PERK inhibition.

We reveal a strong reduction in total P-Tyr levels in response to ER stress when PERK is inhibited, suggesting a defect in upstream signaling. In agreement with this observation, the tyrosine kinase receptor KIT is retained in the ER and accordingly depleted from the cell surface. This phenomenon is not restricted to BRAF mutated cells, but generally affects the ER to surface trafficking of additional key receptor tyrosine kinases (RTKs), such as c-MET and EGFR. Important to cancer therapy, we highlight clinically approved drugs at their pharmacological concentrations, previously reported to predispose mild ER stress, to convey ER retention of these critical oncoproteins. This study describes an alternative mechanism to curtail RTK signaling involving ER retention with cancer therapy implications.

## Results

**The PERK pathway is required to maintain P-Tyr levels under ER stress.** Phosphorylation of tyrosine residues serves mainly as central proproliferation and prosurvival posttranslational modifications in mammalian cells. This signal is particularly critical for cancer cells and is, thus, subjected to pharmacological interventions by kinase inhibitors. To explore whether PERK inhibition is connected to P-Tyr homeostasis, Mel526 cells were treated with the ER stress inducer thapsigargin (Tg), the PERK inhibitor GSK414, ISRIB, or combinations of Tg and GSK414 or ISRIB. P-Tyr levels were strongly reduced only following combined treatments (Fig. 1a, b). These data indicate that UPR/ISR responses buffer P-Tyr levels in response to ER stress.

**PERK or ISR inhibition confers ER retention of KIT under ER stress conditions.** P-Tyr is generated directly by RTKs and cytosolic tyrosine kinases, or indirectly by other types of receptors, such as cytokine receptors. We reasoned that ER stress may interfere with receptor functions, and therefore attenuate the generation of P-Tyrs. Thus, we followed the surface displacement of RTKs. In previous studies we found that MEL526 express high levels of the KIT RTK[10]. This receptor is activated by stem cell factor (SCF), which is present at low levels in the fetal calf serum and constitutively contributes to cellular P-Tyr levels. SDS-PAGE of KIT under reducing conditions resolves two distinct polypeptides; the lower one is ER-resident KIT decorated with high-mannose glycans, and above it is the mature, post ER protein linked to complex glycans. Treatment with Tg alone did not affect the ratio between the mature to immature KIT (Fig. 1c). Under tunicamycin (Tm) treatment, which inhibits N-linked glycosylation, most of KIT was unglycosylated. A portion of it matured normally, suggesting a partial block of glycosylation by Tm. GSK414 alone reduced the levels of KIT due to acceleration of its lysosomal degradation in a PERK-independent manner[10]. However, when Tg or Tm was combined with ISRIB or the GSK414, the intracellular levels of KIT increased at the expense of the mature form. The effect of the inhibitors on PERK verified that GSK414 inhibits the autophosphorylation of PERK, while ISRIB does not (Fig. 1c). This was associated with a strong downregulation of KIT from the cell surface as measured by flow cytometry (Fig. 1d). Of note, ISRIB effect was milder than GSK414, a phenomenon that is attributed to its rheostat mode of action[11].

We then compared the effect of the combined Tg/GSK414 and Tg/ISRIB on the phosphorylation status of KIT in comparison with dasatinib, a multikinase inhibitor used in clinics for the treatment of KIT driven malignancies[12]. This experiment was performed in the presence and absence of the KIT ligand, SCF. As expected, dasatinib prevented the phosphorylation of KIT and reduced total P-Tyr levels without affecting the maturation of KIT. Tg/GSK414, and to a lesser extent Tg/ISRIB, had a similar effect accompanied by a block in the maturation of the protein (Fig. 1e), suggesting that this block is the cause of KIT inhibition rather than the other way around.

GSK414 is not specific to PERK and possesses off target activities[10,13]. We generated PERK KO Mel526 by CRISPR/Cas9 editing to ensure that the delay in KIT maturation is not due to off-target effects. Treatment of the PERK KO cells with Tg or Tm was sufficient to confer the intracellular accumulation of KIT (Fig. 1f compare lanes 3 with 9). Of the three UPR arms, this effect was mediated only by PERK. Deletion of IRE1 in the MEL526 cells did not affect the trafficking of KIT in response to Tg alone. As in WT cells, GSK414 was required to arrest KIT in the ER (Supplementary Fig. 1). To assess the potential contribution of ATF6, we treated the cells with the serine

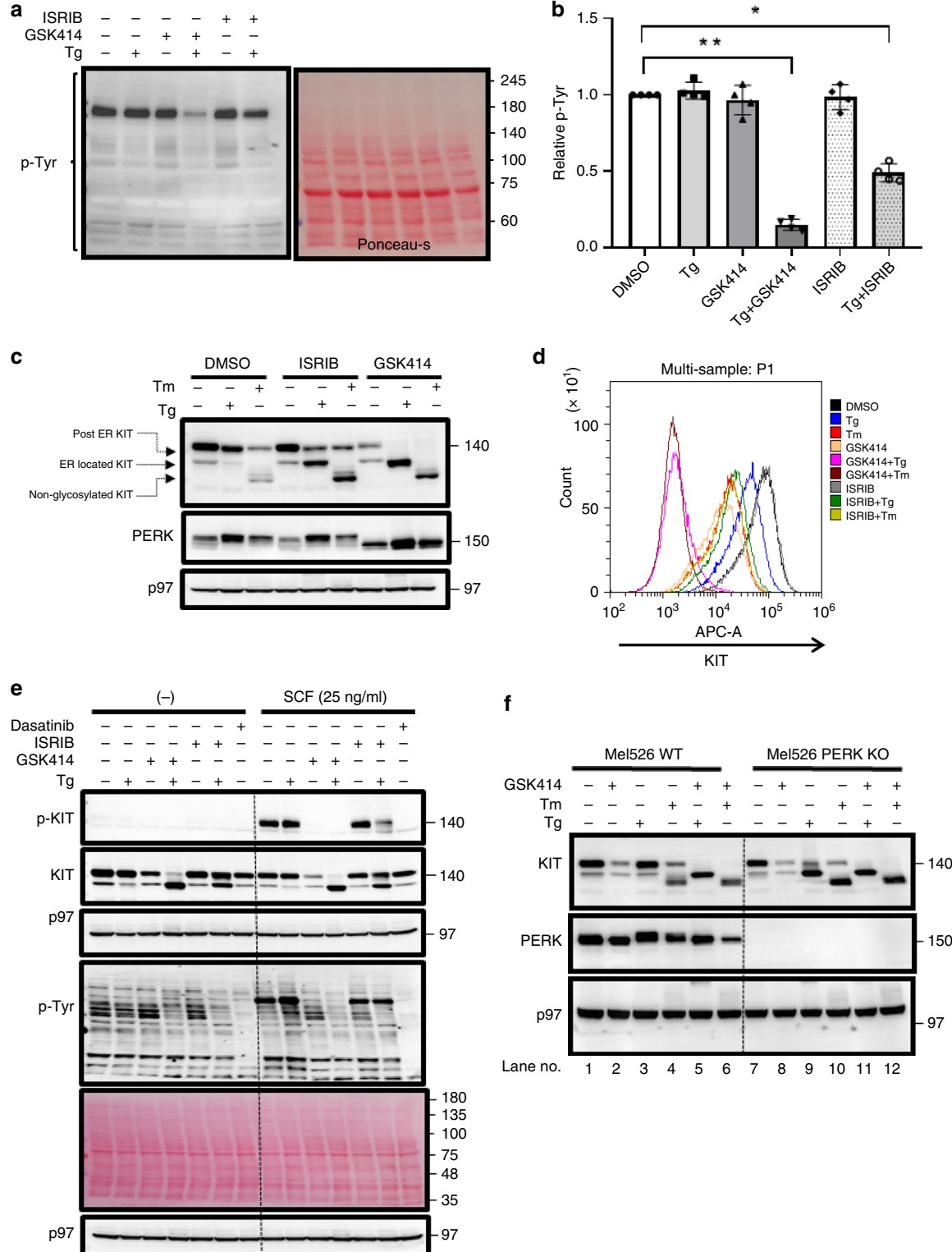

**Fig. 1 Egress from the ER of KIT requires PERK/ISR signaling under ER stress. a** Immunoblotting against p-Tyr and ponceau-s staining of Mel526 cells treated either with DMSO, GSK414 (0.5 μM), ISRIB (0.2 μM), and Tg (0.2 μg/ml) alone or in combinations for 16 h. **b** Quantification relative to control of total P-Tyr levels for each treatment. Ponceau-s staining was used as the loading normalizer (Shown is the average ± SD of four independent experiments average, *$p < 0.05$, **$p < 0.01$, Kruskal–Wallis one-way analysis of variance). **c** Immunoblotting against KIT, PERK, and αβ tubulin of Mel526 cells under Tg or Tm (added at 0.2 μg/ml) induced ER stress in the presence of DMSO, ISRIB, or GSK414 for 16 h. Shown is a typical outcome from three independent repetitions. **d** Corresponding flow cytometry analysis of cell-surface KIT. **e** Mel526 WT cells were treated with Tg (0.2 μg /ml), GSK414 (0.5 μM), ISRIB (0.2 μM), and dasatinib (0.5 μM) alone or in combination. After 16 h, cells were stimulated with SCF (25 ng/ml) for 20 min followed by immunoblotting against p-KIT, KIT, p-Tyr, p97, and ponceau-s. A result of two repetitions. **f** Immunoblotting for KIT, PERK, and p97 of Mel526 WT or PERK KO cells treated with Tg, Tm, GSK414 alone or in combinations for 16 h. Shown is the typical outcome of the experiment ($n = 4$ independent repetitions).

protease inhibitor, pefabloc, that inhibits ATF6 proteolytic cleavage in the Golgi[14]. Inhibition of the cleavage did not affect the ER retention of KIT (Supplementary Fig. 1).

The retention of KIT in the ER following a combined Tg/GSK414 treatment was further validated by EndoH digestion and fluorescence microscopy, using calnexin as an ER marker (Supplementary Fig. 2). The effect on KIT trafficking was specific to ER stress, as other stress conditions (e.g., heat shock, hypoxia, or serum starvation) did not affect the maturation of the KIT receptor by ISRIB alone. Only the Tg/ISRIB combination affected KIT maturation (Supplementary Fig. 3). These data indicate that the PERK/ISR pathway under ER stress conditions is required for the proper egress of KIT from the ER.

To ensure that the ER retention of KIT is not an idiosyncrasy of the Mel526 cells, we expressed KIT in Mel624 and 293T cells that do not express KIT endogenously. We followed its maturation by immunoblotting. In both cell lines the treatment of Tg plus GSK414 resulted in the ER retention of KIT (Supplementary Fig. 4). Again, in PERK KO Mel624, Tg alone was sufficient to confer the ER retention. We also followed the maturation of endogenous KIT in the mast cell leukemia HMC-1.1 cell line. KIT was retained in the ER, similar to what was observed for the other cells (Supplementary Fig. 4). Collectively, PERK is responsible for the ER egress of KIT in the course of ER stress.

**The ER retention is selective**. Next, we wanted to examine if the trafficking of additional RTKs and other cellular glycoproteins is blocked by the treatment. c-MET is an RTK that undergoes proteolytic processing in the Golgi. Thus, a polypeptide with higher molecular weight is indicative of ER localization. c-MET has strong oncogenic properties in hepatocellular carcinoma (HCC), glioblastoma, and other malignancies, c-MET inhibitors were, therefore, suggested for cancer therapy. To date, there is no specific inhibitor to c-MET in clinical use. Those inhibitors that block c-MET, such as cabozantinib, also inhibit other tyrosine kinases and, thus, are highly toxic[15].

We followed the maturation of c-MET in the HCC cell line HepG2. Under the combined Tg/ISRIB or Tg/GSK414 treatments, c-MET pro-form accumulated, indicative of ER retention (Fig. 2a). As was shown for KIT, in HepG2 cells that lack PERK, Tg alone was sufficient to confer the c-MET ER retention (Fig. 2b). This indicates that PERK controls the ER egress of several proteins. We extended the analysis to additional surface proteins. Biochemical analysis showed that EGFR (Fig. 2a) maturation was partially blocked by Tg/GSK414 treatment. Under Tg/GSK414 or Tg/ISRIB treatment the ER retained EGFR underwent additional mannose trimming that accounted for its lower molecular weight. While the surface expression of these RTKs was reduced, expression of HLA-A2 protein was less affected by the same conditions (Fig. 2c). The maturation of the secreted protein α1 antitrypsin (decorated with four N-linked glycans) was unperturbed (Supplementary Fig. 5). Using lectins that recognize either mannosylated proteins (ConA), enriched in the ER or sialylated proteins (WGA) enriched at the cell surface, we did not detect strong effects for the Tg/ISRIB treatment (Supplementary Fig. 5). This suggests that the block in trafficking is not general and selectively affects certain glycoproteins.

To characterize the repertoire of the proteins sensitive to this pharmacological treatment, surface proteins were isolated from HepG2 cells treated with Tg alone (control) or Tg/GSK414 combination. The assumption was that surface downregulated proteins are subjected to selective ER retention (sERr). Mass spectrometry analysis of the extracted proteins identified 141 proteins that were significantly downregulated upon the Tg/GSK414 treatment. Most were not membrane integral proteins,

but rather membrane associated (Fig. 2d in blue, Supplementary Data File 2). This subset of proteins includes c-MET and EGFR but not HLA-A2, pointing to the selectivity of the Tg/GSK414 treatment (Fig. 2d). The proteomic results for c-MET and HLA-A2 were validated biochemically in the total lysates compared with the surface protein fraction. c-MET levels decrease in the surface protein fraction, while the heavy chain levels of HLA-A2 were unaltered upon the Tg/GSK414 treatment (Supplementary Fig. 5). Functional annotation of the differentially downregulated proteins (in blue) revealed that the downregulated proteome is significantly enriched in proteins with kinase activity (Fig. 2e). We, therefore, termed this phenomenon as sERr.

**sERr is associated with formation of high molecular weight complexes held by disulfide bonds**. We analyzed the requirement of ATF4 that executes the main transcription programs downstream to PERK. We examined the requirement of ATF4 signaling for sERr. If lack of ATF4 under stress enforces sERr, then ATF4 overexpression should rescue trafficking to the cell surface. HepG2 cells were transfected with increasing amounts of a human ATF4-encoding plasmid, and the maturation of c-MET was assessed under Tg/ISRIB conditions. The expression of ATF4 did not rescue the trafficking of c-MET, suggesting that this transcription factor is not controlling sERr (Supplementary Fig. 5).

We then addressed the translation regulation by PERK. Inclusion of the PERK inhibitor under stress conditions promotes overall protein synthesis to further explore potential mechanisms that control sERr[16]. If the unbalanced synthesis is the reason for sERr, then attenuation of protein synthesis should rescue trafficking to the cell surface. We subjected HepG2 cells to the protein synthesis inhibitor cycloheximide (CHX) and assessed sERr by the ratio of pro-MET to its mature form. The magnitude of sERr was reduced as CHX concentration was increased. This was supported by a small increase of c-MET and EGFR levels at the cell surface, albeit the reduction in their synthesis (Fig. 3a, b). Thus, the basis of sERr is the misbalanced protein synthesis conferred by the PERK inhibitor.

A recent study shows that the amyloidogenic protein, transthyretin, undergoes aggregation under Tg plus PERK inhibition[17]. We blotted the cell extracts under nonreducing conditions to investigate whether sERr is also associated with formation of aggregates. For both KIT and c-MET we observed the generation of large molecular weight complexes that were sensitive to reducing conditions (Fig. 3c).

ERp44 is a PDI protein that participates in the ER retrieval and retention of prominent ER proteins, such as Ero1α[18], FGE[19], and ERAP1[20] by forming mixed disulfide bonds. When ERp44 was examined by immunoblotting under nonreducing conditions, it aggregated into DTT-sensitive high molecular weight complexes only under sERr conditions, similar to the RTKs (Fig. 3d). ERp44 undergoes O-glycosylation in the Golgi, which yields a band slightly higher than the nonmodified ERp44 on SDS-PAGE[21]. Tg treatment alone or together with the PERK inhibitor enhanced the proportion of glycosylated ERp44, indicating a trafficking through the Golgi. KIT-deficient Mel526 cells were reconstituted with triple Flag-tagged KIT to study the interaction of KIT and ERp44. Immunoprecipitation of the FLAG containing KIT protein under control or Tg/GSK414 conditions showed that the glycosylated ERp44 preferentially interacts with the ER retained KIT (Fig. 3e). Analysis of total PDI proteins did not indicate a similar aggregation pattern (Fig. 3d), which may be a limitation of the antibody. We did observe enhanced interaction of KIT with PDI under sERr conditions (Fig. 3f). This indicates that sERr is probably a consequence of a misfolding processes related to

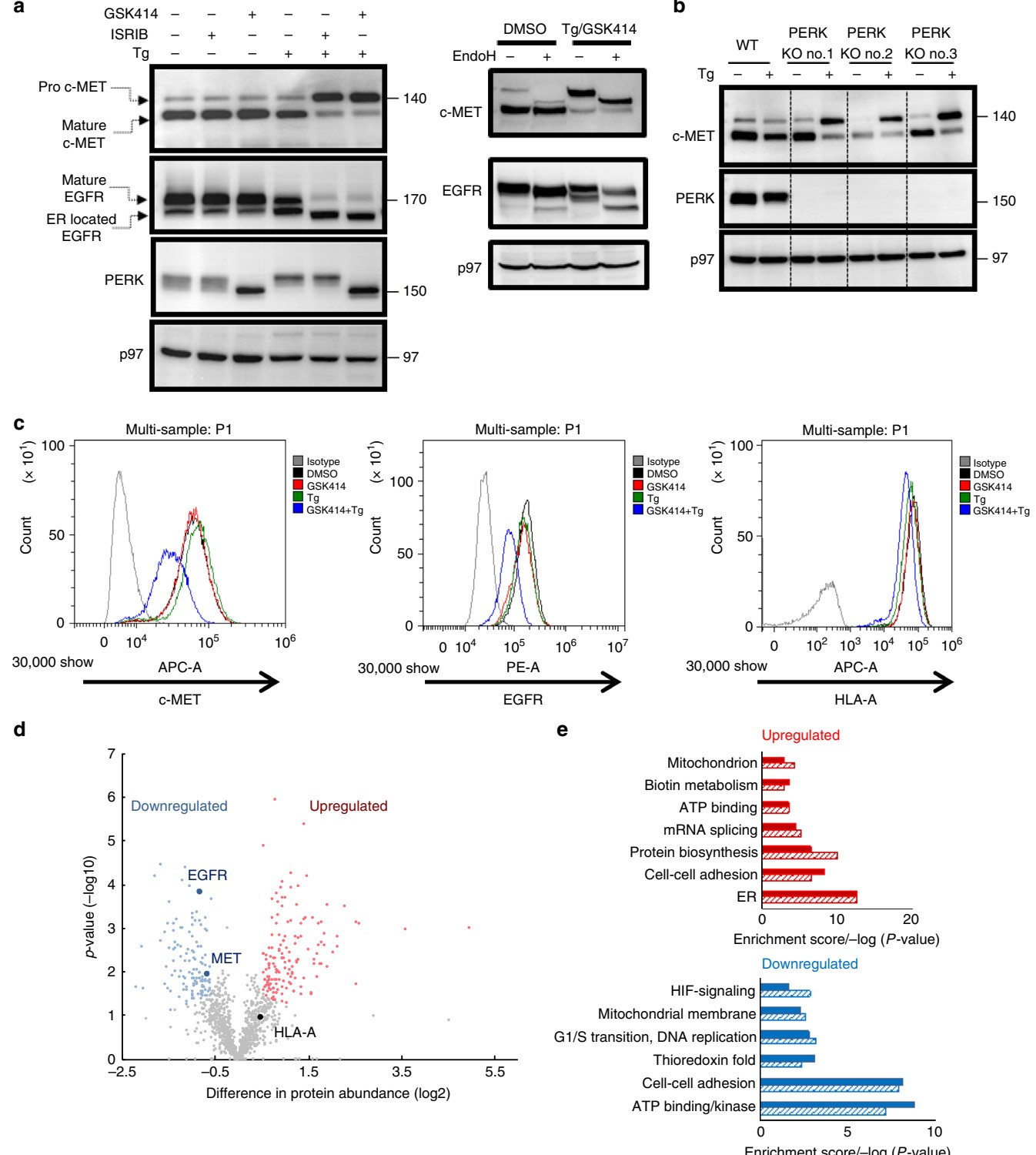

**Fig. 2 Impairment of PERK/ISR signaling under ER stress confers the ER retention of a subset of proteins, including prominent oncogenic RTKs.**
**a** Immunoblotting and ponceau-s staining as for c-MET, EGFR, PERK, and p97 as loading control in HepG2 cells treated with Tg, GSK414, and ISRIB alone or in combinations for 16 h (typical result of three independent repetitions). **b** Immunoblotting for c-MET, PERK, and p97 in WT HepG2 and in three different clones of PERK KO HepG2 cells, treated either with DMSO or Tg alone for 16 h; **c** Flow cytometry analysis of surface c-MET, EGFR, and HLA-A in WT HepG2 treated with DMSO, Tg, and GSK414 for 16 h (typical result of three independent repetitions). **d, e** Comparative proteomics of the surface proteins upon treatment with Tg in absence and presence of GSK414. The differently expressed proteins shown by the volcano plot (**d**). Significantly upregulated proteins upon the Tg/GSK414 treatment are in red, the downregulated proteins are in blue, according to the FDR of 0.05 and a fold change greater than two. The volcano plot is related to Supplementary Data Files 1, 2. **e** Functional enrichment analysis of the upregulated and downregulated proteins affected by the Tg/GSK414 treatment. The enrichment score is above 1.7 (solid bars) with the Benjamini p value < 0.005 (dashed bars).

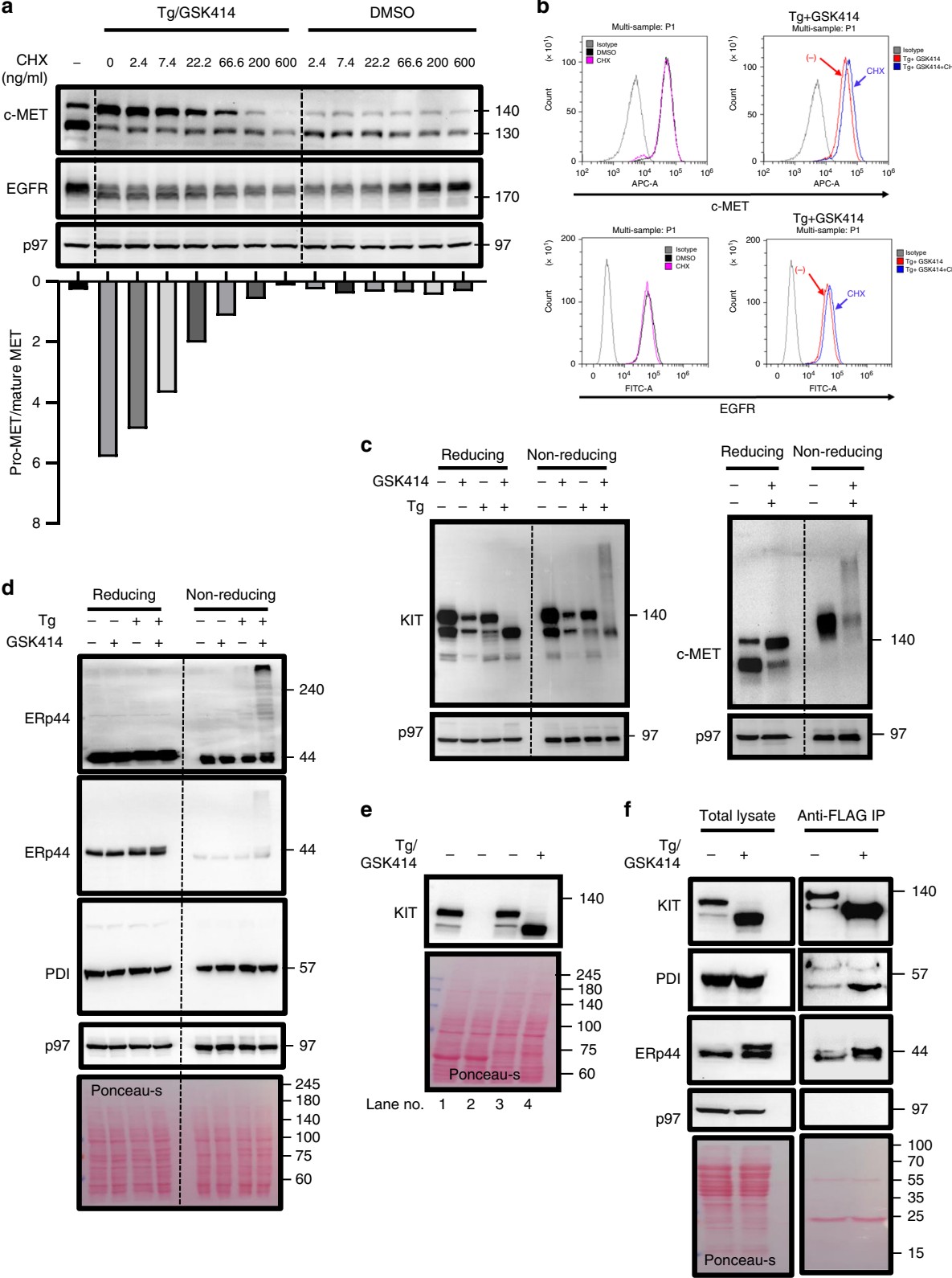

disulfide bond isomerization, in which ERp44 plays an important role. To test whether ERp44 is required for sERr, we knocked it out from HepG2 cells, and we followed the sERr of c-MET. While the absence of ERp44 did not prevent sERr, its efficiency was diminished as determined by the ratio of pro-c-MET to its mature form. We also observed a reduction in the molecular weight of the ER retained nonreduced complexes, demonstrating that ERp44 promotes the intermolecular disulfide bond binding of sERr clients (Supplementary Fig. 6A, B).

**sERr is induced by HIV protease inhibitors in combination with ISRIB as a potential anticancer combination.** The effect of sERr on major proto-oncogenes, such as c-MET and EGFR,

**Fig. 3 sERr is attenuated by translation inhibition and involves the generation of intermolecular disulfide bonds formation with PDI proteins. a** HepG2 WT cells were treated with either DMSO or Tg (0.2 μg/ml) plus GSK414 (0.5 μM) in the presence of different concentrations of cycloheximide (CHX) for 16 h followed by immunoblotting against c-MET, EGFR, and p97. The quantitative ratio between ER-arrested c-MET to cell-surface located c-MET is shown as pro-MET/mature MET for each treatment. Experiment was performed twice with a similar outcome. **b** Flow cytometry analysis of EGFR and c-Met of HepG2 WT cells following 16 h treatment either with DMSO in presence/absence of CHX (600 ng/ml) or with Tg(0.2 μg/ml) plus GSK414(0.5 μM) in presence/absence of CHX(600 ng/ml). **c** Reducing and nonreducing immunoblotting for KIT in Mel526 cells (left) and c-MET in HepG2 cells (right) treated with DMSO, Tg, GSK414 alone or with Tg/GSK414 combination (typical results of three repetitions). **d** Reducing and nonreducing immunoblotting for ERp44 and PDI of HepG2 cells treated either with DMSO, Tg, GSK414, or both for 16 h (typical results of three repetitions). **e** Mel526 KIT KO cells were generated by CRISPR/Cas9 gene editing (lane 1 is WT cells where lane 2 is a KIT KO). The fusion protein KIT-3xFLAG was then stably expressed in the KO cells (lane 3). ER retention of the fusion protein was validated by treatment with Tg + GSK414 (lane 4). Mel526 KIT-3xFLAG expressing cells were treated either with DMSO or with Tg/GSK414 for 16 h. **f** Total lysate or anti-FLAG immunoprecipitates were immunoblotted for KIT, PDI, and ERp44 with ponceau-s staining(right) (typical results of three repetitions).

invokes a mechanism for cancer treatment. It is postulated that in addition to the simultaneous prevention of several kinase receptors from reaching the cell surface and encountering their cognate ligand, the sERr should elicit a terminal UPR, owing to misfolding of ER client proteins. We, therefore, sought to find sERr-inducing drug combinations amenable for cancer therapy. GSK414 is most likely not suitable for clinical use owing to weight loss and hyperglycemia. ISRIB, on the other hand, has shown less toxicity and thus is proposed to have a better potential for clinical use. Versions of ISRIB with improved pharmacokinetic properties are currently being investigated[8]. As shown, an insult to the ER itself is necessary to elicit the sERr. Since mild concentrations of Tg were sufficient to confer retention of c-MET (Supplementary Fig. 7), we reasoned that clinically approved drugs that elicit mild ER stress can replace Tg.

Nelfinavir and lopinavir are HIV protease inhibitors that also induce ER stress and promote eIF2α phosphorylation[22,23]. These drugs are particularly effective to induce ER stress in the liver[24]. We evaluated the effects of nelfinavir and lopinavir, when combined with ISRIB, to affect the trafficking of RTKs to the cell surface in HepG2 cells. The peak serum concentration of these drugs in AIDS patients at their clinical dosing regimen is in the low millimolar range[25,26]. Both drugs elicited sERr of EGFR and c-MET in vitro at concentrations 100-fold lower than their serum $C_{max}$ (Fig. 4a, b). Using anti-human serum antibodies, we have not observed a block in secretion under nelfinavir/ISRIB or lopinavir/ISRIB sERr treatments (Supplementary Fig. 8). Thus, the effect of these sERr combinations may open a therapeutic window to affect RTK-addicted tumors with minimal toxicity to normal liver cells.

We analyzed the survival and tyrosine phosphorylation in two RTK-addicted cells to assess the therapeutic potential of lopinavir and nelfinavir when combined with a PERK inhibitor. HCC827 are lung adenocarcinoma cells that have acquired mutation in an EGFR tyrosine kinase domain. These cells succumb to the EGFR inhibitor, osimertinib. Analysis of the phosphorylation status of EGFR and c-MET in these cells indicates a full inhibition of their phosphorylation by osimeritinib. A similar inhibition was observed only when nelfinavir or lopinavir were combined with the PERK inhibitor, accompanied by their ER retention (Fig. 4c). This was reflected in the total P-Tyr levels and in the viability of the cells, determined by their light scattering properties (live cells shown in red, Fig. 4d). The mast cell leukemia cell line, HMC-1.1, was addicted to KIT that was blocked by dasatinib. Since GSK414 enhanced the degradation of KIT in an off-target manner, we used a different PERK inhibitor, AMG PERK 44, that does not inhibit KIT and does not affect its turnover[10]. Consistent with the data in Fig. 4c, lopinavir and nelfinavir when combined with AMG PERK 44 conferred sERr, reduced the total P-Tyr levels, and inhibited KIT phosphorylation (Fig. 4e). As expected, this was reflected in HMC-1.1 survival (Fig. 4f). These data strongly

suggest that an sERr cocktail can be used to treat RTK-addicted tumors.

To explore this possibility further, we compared sERr efficiency in HepG2 cells with the hTERT cell line established from normal human hepatocytes immortalized by telomerase[27]. Total P-Tyr levels were strongly reduced in HepG2 cells but not in hTERT cells under the sERr conditions (Fig. 5a, lanes 5 and 9). This coincided with elevated toxicity and impact on proliferation for HepG2 cells (Fig. 5b). After 4 days of drug treatment only the combined treatment resulted in few live cells, while each drug alone was less toxic (Supplementary Fig. 9). Taken together, similar to TKIs, sERr-induced therapy may have selectivity to tumors.

**The combination of nelfinavir and ISRIB is effective to inhibit HepG2 xenografts without evidence of liver toxicity.** We then examined the potential of sERr in human xenografts. NOD/SCID mice were inoculated subcutaneously with HepG2 cells. In this model a palpable tumor is formed within a week, and within two weeks the tumor reaches a volume of more than 1 cm³, which requires euthanasia. We challenged the mice with HepG2 cells and started the treatment three days later, before signs of tumor growth to ensure proper randomization. Mice were treated with a daily injection of vehicle, ISRIB, lopinavir, nelfinavir, and combinations of lopinavir/ISRIB and nelfinavir/ISRIB for 14 days. Only the combination of nelfinavir/ISRIB significantly inhibited tumor growth. In half of the mice HepG2 tumors did not develop at all or were barely seen under nelfinavir/ISRIB combination. All the other treatments were equivalent to the vehicle control (Fig. 6a). The lack of lopinavir/ISRIB efficacy may be related to its rapid metabolism by the cytochrome p450 enzymes[28]. Addition of ritonavir, which serves in the clinics as a cytochrome inhibitor in combination with lopinavir, may also help to promote sERr with this drug. Examination of liver damage, assessed histologically and by serum AST/ALT transaminase measurements, indicated no evidence of it (Supplementary Fig. 10). Hence, sERr treatment by nelfinavir/ISRIB had a strong anticancer effect with no observed hepatotoxicity within this time frame.

We repeated the experiment to validate that the treatment affects the expression and trafficking of c-MET and EGFR in vivo. This time, however, tumors were allowed to develop for 10 days in the NOD/SCID mice without treatment. Then, mice were treated for 3 days with vehicle, nelfinavir, or the nelfinavir/ISRIB combination, and tumors were excised. Western analysis of total protein extracts showed a significant reduction in total P-Tyr levels (Fig. 6b). Remarkably, Immunohistochemistry (IHC) for total P-Tyr revealed in the combined treatment, regions that are devoid of P-Tyr. Staining for human α1-antitrypsin was homogenous (Supplementary Fig. 10). Immunoblotting of bulk tumor lysates showed that maturation of c-MET was not fully blocked. However, a clear accumulation of immature pro-c-MET

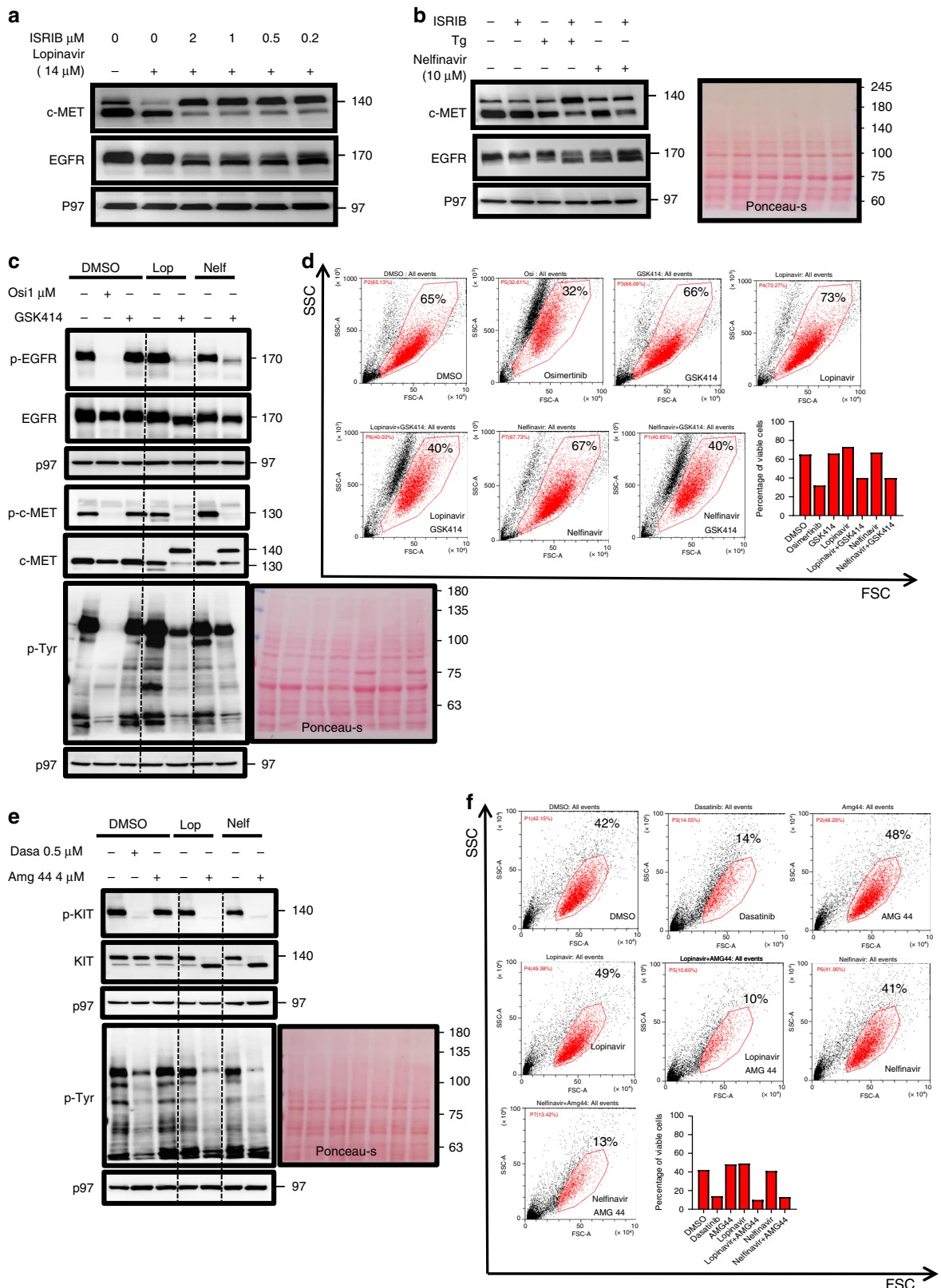

was observed in the combined nelfinavir/ISRIB-treated group, indicating ER retention of c-MET in vivo. ER-localized immature and mature EGFR were not clearly resolved by SDS-PAGE, perhaps owing to contribution of the mouse EGFR signal derived from the mouse tumor stroma (Fig. 6b). We conclude that

pharmacological induction of sERr by nelfinavir/ISRIB treatment is effective in vivo to curtail the total output of RTKs and can be used for cancer therapy. Both drugs, however, must be simultaneously delivered to the tumor to elicit effective reduction in P-Tyr levels and compromise RTK trafficking.

**Fig. 4 The combination between lopinavir and nelfinavir with PERK inhibitors elicits sERr, reduces total P-Tyr levels and compromises viability of RTK-addicted cells similar to TKIs. a** Immunoblotting against c-MET, EGFR, and p97 and ponceau-s staining of HepG2 cells treated with Tg, nelfinavir (20 µM), and ISRIB alone or in combinations for 18 h. **b** Immunoblotting against c-MET, EGFR, and p97 for HepG2 treated with DMSO or lopinavir (20 µM) with different concentrations of ISRIB for 18 h; **c** HCC827 WT cells where treated with osimertinib (1 µM), GSK515 (0.5 µM), lopinavir (26 µM), and nelfinavir (16 µM) alone in combination for 32 h. Cells where then analyzed by immunoblotting for p-EGFR, EGPR, p-MET, c-MET, p-Tyr, p97. Shown is one of two independent repetitions. **d** Cells were analyzed by flow cytometry to determine viability (shown in red). Bars represent the percentage of live cells. **e** HMC-1.1 cells where treated with dasatinib (0.5 µM), GSK515 (0.5 µM), lopinavir (26 µM), and nelfinavir (16 µM) alone in combination for 32 h. Cells where then analyzed by immunoblotting for p-KIT, KIT, p-Tyr, p97. Shown is one of two repetitions. **f** Cells were analyzed by flow cytometry to determine viability (shown in red). Bars represent the percentage of live cells.

## Discussion

sERr was first described in yeast subjected to reductive agents, such as DTT. Selectivity emphasized proteins with large number of disulfide bonds, while proteins with free sulfhydryl moieties were secreted normally[29]. While this study demonstrates retention under extreme reductive conditions, subtle perturbation of ER redox can affect trafficking. As such, the mere overexpression of PDI is sufficient to retain proteins rich in disulfide bonds in the ER[30]. Here we demonstrate the induction of ER retention with drugs at their pharmacological range with a clear potential for cancer therapy.

Cancer therapy has been transformed in recent years owing to the power of genomics and the expansion of therapeutic options. A growing arsenal of kinase inhibitors is now available and can be tailored to the genetic makeup of the tumor, a hallmark of personalized therapy. While this strategy may have obvious advantages over nondiscriminatory classical chemotherapy, resistance to kinase inhibitors almost inevitably develops, greatly challenging and complicating clinical practice. The ability to curtail multiple RTKs outputs without using a large number of drugs may have important therapeutic advantages. One of the mechanisms of resistance to TKIs is the establishment of missense point mutations that diminish drug binding. This has been documented for KIT[31], PDGFRA[32], EGFR[33] as well as for serine/threonine kinases downstream to the RTKs. While these mutations typically pose a bad prognosis for the patient owing to their tremendous effect on TKI affinity, they are not expected to override sERr, as they do not affect luminal cysteine residues or kinase folding. Thus, sERr can be used as a supplemental treatment to TKIs, which may improve outcome, reduce the probability of establishing resistance by the tumor, and perhaps lower the TKI dose.

The PERK pathway simultaneously controls prodeath[34] and prosurvival[35] mechanisms. Thus, its role in cancer cannot be predicted and needs to be tested in the relevant tumor/therapy context. Lack of PERK in humans, called the Wolcott-Rallison syndrome, is characterized by permanent neonatal diabetes mellitus, bone defects, and episodes of acute liver failure. Thus, long term therapy with PERK inhibitors is associated with severe metabolic toxicities. The dichotomy of PERK in life/death decisions is also manifested in metabolism. Liver steatosis following acute ER stress has been largely attributed to PERK activation[36]. We, thus, propose that PERK inhibition can be well tolerated by the normal liver parenchyma for short-term cancer therapy. For this and other reasons HCC can serve as good model to test the sERr approach with PERK inhibitors, owing to the relative ease to target drugs to the liver and the fact that HCC tumors develop addiction to RTK signaling.

What function of PERK inhibition promotes sERr? We can exclude inhibition of ATF4 production, since heterologous ATF4 overexpression does not affect the process of sERr. Reduction of ATF4 production should also result in attenuation of proapoptotic CHOP expression, which would not correlate with compromised growth of HepG2 cells and diminished tumor size

caused by SERR induction. Our combined data obtained using GSK414 and ISRIB point to a central role of releasing translational inhibition during ISR, since both GSK414 and ISRIB re-induce translation by means of activation of the ternary initiation complex consisting of eIF2α, GTP, and tRNA-Met. Thus, the ER stressors, Tg and Tm, cause protein accumulation in the ER due to misfolding and an imbalance of misfolded proteins/aggregates and chaperones. GSK414 or ISRIB treatment will further force an increase in novel ER proteins, again resulting in even more unfolded and aggregated proteins. The resulting high protein density and close proximity of misfolded proteins might then allow for the formation of unusual intermolecular disulfide bridges.

Since PERK is only one of three stress sensors of the UPR, it leaves the question if the other sensors, IRE1 and ATF6, contribute to the process of sERr? The fact that we did not observe a qualitative difference in sERr in IRE1α-deficient cells, or when ATF6 activation was inhibited, does not exclude a promoting role of IRE1α. Wang et al. have demonstrated a positive role of IRE1α for the expression of PDI[37], which might again promote the formation of intermolecular disulfide bridges. In addition, PDI enzymes were shown to induce PERK activation[38], which should be impaired in the presence of GSK414 or ISRIB.

Several key questions still remain unanswered. What are the structural features that subject a glycoprotein to sERr? The proteomic analyses identified a small subset of proteins to undergo downregulation from the surface under sERr conditions. The common denominator of these proteins is unknown and may relate to size, number of disulfide bonds, and/or specific interaction with certain PDI proteins. ERp44 definitely plays a role, but probably in conjunction with other players. Exploration of the early normal protein folding of sERr clients may shed light on its selectivity.

What processes does PERK regulate to prevent sERr under ER stress? Attenuation of protein synthesis allowed the ER to recover under sERr therapy and restore trafficking. This suggests that retention and misfolding are primarily a result of overload in the ER with respect to the task of disulfide bond formation. In addition to the overload per se, PERK can regulate local pH, calcium concentrations, and chaperone expression, thus facilitating folding. Identification of the factors that facilitate the folding of KIT or c-MET, and are compromised when PERK is inhibited is needed. This analysis is critical to predict mechanisms of resistance to sERr that are likely to develop. A possible mechanism relates to protein synthesis, reverting the cells into eIF4E-independent translation as recently demonstrated for MEFs[39].

Lastly, the role of sERr in cancer therapy remains to be determined, particularly with respect to a combination with kinase inhibitors. The potential contribution of sERr to the development of resistance to TKIs must be assessed, owing to orthogonality of these two pharmacological approaches in curtailing TKI signaling. The development of resistance to sERr, itself, should be studied. Plausible mechanisms of resistance to

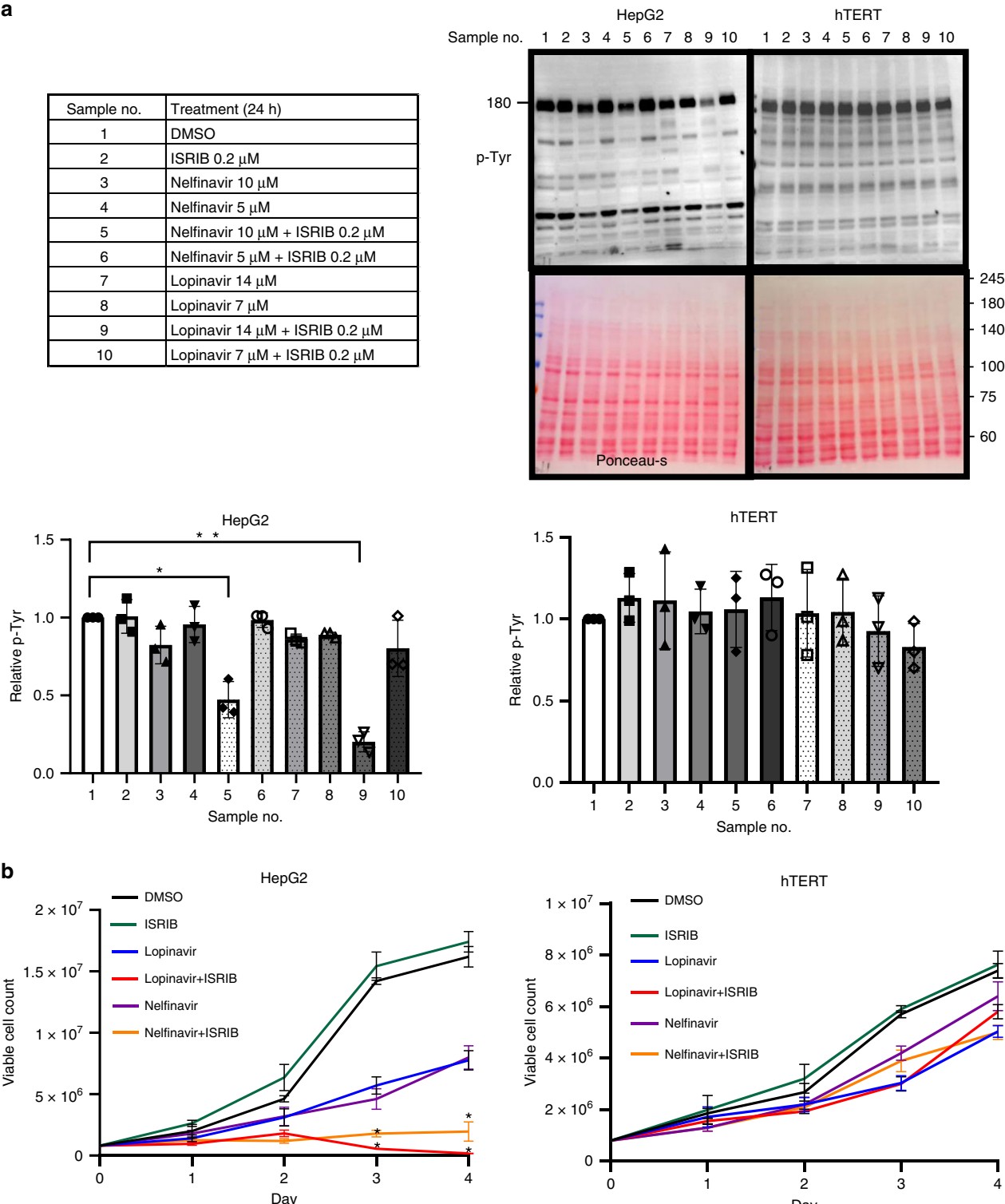

**Fig. 5 The combination between lopinavir or nelfinavir with ISRIB elicits sERr, reduces total P-Tyr levels and compromises growth of HepG2, less of hTERT cells. a** Immunoblotting for P-Tyr and ponceau-s staining of HepG2 cells and of hTERT treated with DMSO, nelfinavir, lopinavir, and ISRIB as specified in the table (left). Quantification of total P-Tyr relative to ponceau-s is shown in bar graphs (right) for both experiments. Average of three independent experiments ± SD. *$p > 0.05$, **$p < 0.01$, Kruskal–Wallis one-way analysis of variance. **b** Monitoring of proliferation evaluation of HepG2 and hTERT cells treated with DMSO, ISRIB (0.2 μM), nelfinavir (14 μM) (top-left), and lopinavir (14 μM) (top-right) alone or in combination. Data are shown as viable cell count (VCC) vs. time (days) conducted in triplicates for each treatment. Error bars represent S.E.M. Statistical significance was calculated for each combined treatment relative to drug only. *$p > 0.05$ were calculated using Student's $t$ test (two-tailed).

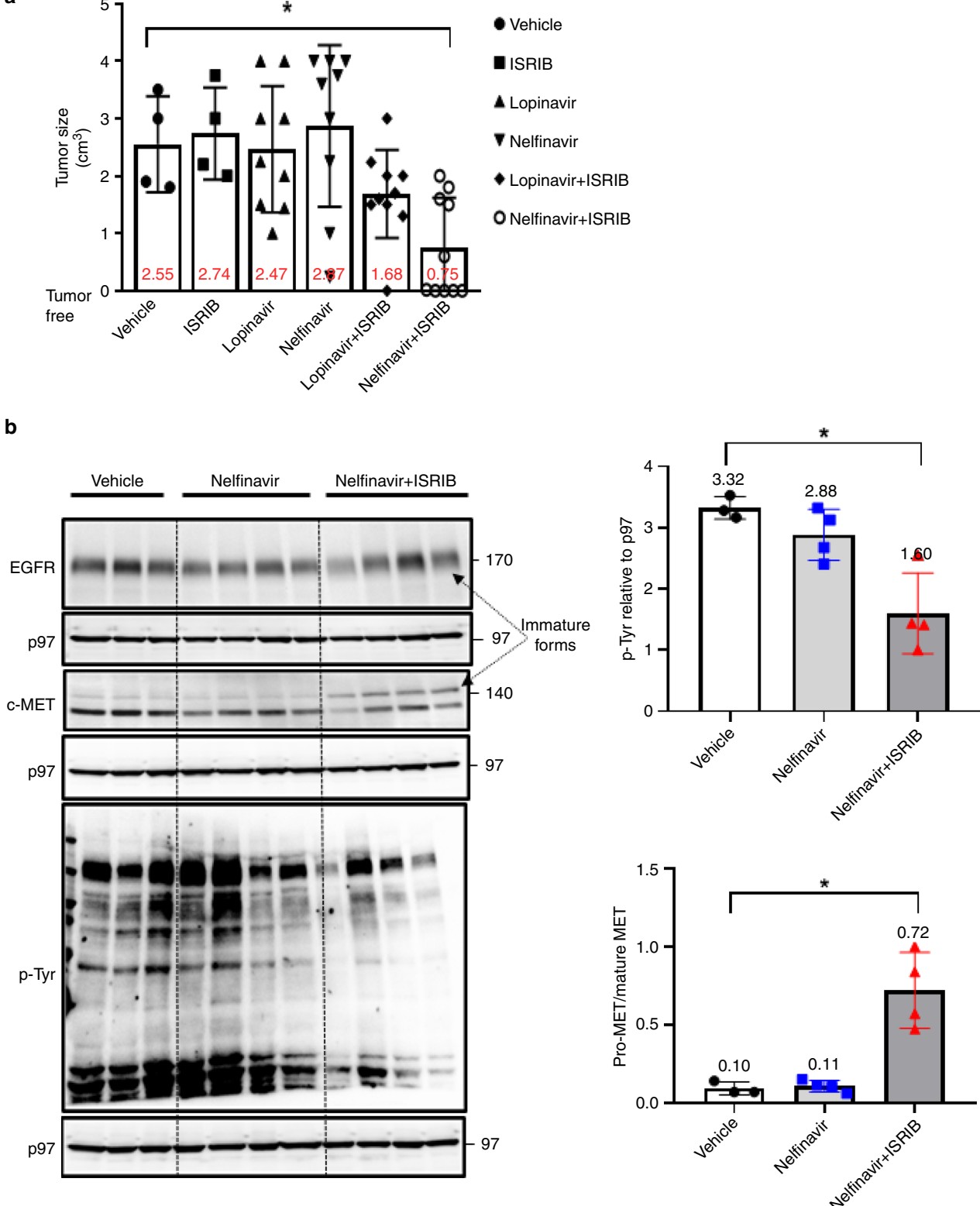

sERr may impinge upon redox potential and protein synthesis, pathways critical for tumor progression in general. The engagement of these mechanisms is important to define which tumors are likely to benefit from sERr. While HCC tumors, particularly those that amplify the c-MET locus, may be a good initial indication for sERr, other tumors respond initially well to TKIs and then develop undruggable resistance. They may also be considered for sERr. This includes several prevalent malignancies, such as breast cancer and glioblastoma. The potential of sERr as an adjuvant is supported by the analyses of the RTK-addicted cells that show similar efficacy to the potent TKI in compromising viability, with the distinction that the oncogenic kinase, itself, is not directly inhibited. Mutation in the kinase domain of the TKIs that have shown to develop in response to TKIs is

**Fig. 6 Nelfinavir in combination with ISRIB significantly diminishes tumor size without causing a noticeable hepatotoxicity. a** NOD-SCID mice were subcutaneously injected with HepG2 cells (two million cells/mouse). Three days following the xenografts injection, mice were treated daily either with vehicle, ISRIB (2.5 mg/kg), lopinavir (17 mg/kg), nelfinavir (50 mg/kg), lopinavir + ISRIB, or nelfinavir + ISRIB for 14 days. Tumors dimensions were measured with a caliper. Data are shown in whisker-diagram as a tumor size (cm³) for each treatment (four mice were used for vehicle and ISRIB groups and nine animals for the other groups). Averages are indicated. Error bars represent SD. $p < 0.05$ was obtained only for nelfinavir/ISRIB treatment by one-way ANOVA test. **b** NOD-SCID mice bearing 10 days old subcutaneous HepG2-derived xenografts were intraperitoneally injected with vehicle, nelfinavir, or nelfinavir/ISRIB twice a day for three consecutive days. Tumor were excised and immunoblotting to EGFR, c-MET, p-Tyr, and p97 was performed (left). Shown is the quantified total p-Tyr levels relative to p97 as loading control for each mouse and the pro-MET to mature MET ratio, indicative for sERr efficiency. $N = 3$ for vehicle and $N = 4$ for nelfinavir or nelfinavir/ISRIB groups. Average values are indicated. Bars represent SD. Error bars represent SD. $*p < 0.05$ by Kruskal–Wallis one-way analysis of variance.

unlikely to affect sERr efficacy, since they occur in the cytoplasmic catalytic domain. We, therefore, think that inclusion of sERr early during treatment in conjunction with the kinase inhibitors should improve efficacy and delay resistance. However, delivery to the tumor environment may be needed either by nanocarriers or simply by direct administration, as feasible for HCC and glioblastoma, to minimize systemic toxicity.

## Methods

**Cell lines and culturing conditions**. Mel526, HepG2, Mel624, hTERT, HCC827, and 293T cells were cultured in high glucose Dulbecco's Modified Eagle medium (DMEM high glucose, Sigma-Aldrich) supplemented with 10% fetal bovine serum (Invitrogen), 2 mM L-glutamine (Biological Industries, Israel), 1% penicillin–streptomycin solution (Biological Industries, Israel), and 1 mM sodium pyruvate (Biological Industries, Israel). HMC-1.1 cells were cultured in Iscove's Modified Dulbecco's Medium (ThermoFisher Scientific) supplemented with 10% calf serum–iron fortified (Sigma-Aldrich) 2 mM L-glutamine, 1% penicillin–streptomycin solution, and 1.2 mM monothioglycerol (Sigma-Aldrich).

**Chemical reagents**. GSK2606414 (TOCRIS #5107), CHX (Sigma-Aldrich #66819), dasatinib (Sigma-Aldrich CDS023389), ISRIB (Sigma-Aldrich SML0843), Tg (abcam #ab120286), osimertinib (LC labs, O-7200), Tm (abcam #ab120296), nelfinavir (Glentham Life Science #GP7332), lopinavir (Sigma-Aldrich #SML1222), sodium (meta)arsenite (Sigma-Aldrich #S7400), and cobalt chloride (Sigma-Aldrich #409332).

**Proliferation kinetic assessment and viability assay**. The same number of cells was plated on 24-well plates. Three wells were harvested on each day of the experiment, and cell counting was performed using hemocytometer and trypan blue exclusion to discriminate live and dead cells. The mean of three wells was recorded as cell number per well or cells per ml medium. On the last day of the experiment (day 4), the cell viability assay was performed using the MTT Cell Proliferation Assay Kit (abcam, #ab211091).

**Generation of knockout cells using CRISPR/Cas9**. The preparation of CRISPR/Cas9 mediated knockouts was performed as previously described according to the Zhang Lab protocols[40]. Transfections were done with the TransIT®-2020 transfection reagent (MirusBio, Madison, WI, USA). Single-cell clones were screened by immunoblotting for the relevant protein. gRNA sequences used for human PERK: 5′-CCGAGGCTCCTGCTCTCCCG-3′; 5′-GAATATACCGAAGTTCAAAG-3′, human IRE1: 5′-CTTTTATGTCTGGCAGCGGG-3′, and human KIT: 5′-GCCTA ATCTCGTCGCCCACG-3′ and 5′-TGAACCACTAGCTTTCCAAA-3′. Human ERp44 5′-GAT GTT GCA TCC AAT TTT TG-3′ and 5′-AAA TTC TTC CTT AAT GAC AT-3′

**Western blotting**. Cells were either trypsinized or directly harvested by cell scraping, centrifuged at $1000 \times g$ for 5 min, and then washed twice in cold PBS. For cell lysis, RIPA buffer supplemented with protease and phosphatase inhibitors was added in a volume of about four times the cells' pellet, then vortexed for 20 min at 4 °C. Lysates were cleared by centrifugation at $12,000 \times g$ for 30 min at 4 °C. 5X reduced Laemmli sample buffer was added, boiled for 5 min at 95 °C, and loaded on SDS-PAGE. The same procedure was performed for nonreducing SDS-PAGE using 5X nonreduced Laemmli (lacking DTT). Following SDS-PAGE, gels were blotted onto PVDF membranes using Biorad PowerPac™. Blots were blocked either with 10% skim milk or in 5% BSA (in case of p-Tyr immunoblotting) both in a TBST buffer at room temperature for 1 h. The following primary antibodies were used: mouse anti p-Tyr(PY99) (Santa Cruz Biotechnology #sc-7020, 1:500), mouse anti-FLAG (M2) antibody (Sigma-Aldrich #F1804, 1:500), rabbit anti-KIT antibody (cell signaling #3074, 1:1000), rabbit anti p-KIT (Tyr719) antibody (cell signaling #3391, 1:1000), rabbit anti-PERK (cell signaling #5683, 1:1000), rabbit anti-IRE1 antibody (cell signaling #3294, 1:1000), rabbit anti-EGFR antibody (cell

signaling #4267, 1:1000), rabbit anti p-EGFR (Tyr1068) antibody (cell signaling #2234, 1:1000), rabbit anti-ERp44 antibody (cell signaling #2886, 1:1000), rabbit anti human serum (Sigma-Aldrich # H3383, 1:50), mouse anti-MET antibody (cell signaling #3127, 1:500), rabbit anti p-MET(Tyr1234/1235) antibody (cell signaling #3077, 1:1000), rabbit anti-PDI antibody (cell signaling #3501, 1:1000),mouse anti ATF6 antibody (Imgenex #IMG-273, 1:250), rabbit anti-α/β tubulin (cell signaling, #2148, 1:1000), anti HIF-1α (D2U3T) rabbit mAb (cell signaling #14179, 1:500), rabbit anti HSP70 (cell signaling #4872, 1:500), rabbit anti LC3B (cell signaling #2775, 1:1000), MICA(2C10) (SANTA CRUZ Biotechnology #sc-23870, 1:250), rabbit anti α1-antitrypsin (abcam #ab129354, 1:500), Peroxidase Wheat Germ Agglutinin (Vector laboratories #PL-1026, 5 µg/ml), Concanavalin A peroxidase conjugate (Sigma-Aldrich #L6397, 1 µg/ml), polyclonal rabbit anti p97 (1:1000), and HC70 (1:1000) were provided by Dr Ariel Stanhill (Open University, Israel). Secondary HRP-conjugated goat anti-rabbit (1:10,000) and anti-mouse (1:10,000) (Jackson Immunoresearch, West Grove, PA) were used. Blots were developed in Bio-Rad ChemiDoc™ XR and analyzed using Image Lab™ software.

**Generation of KIT-3xFlag fusing protein stably expressing cells**. The 3XFLAG epitope was cloned in a frame to the C-terminus of hKIT. hKIT-3xFLAG fusion was cloned into pcDNA 3.1 (+) expression vector between NheI and NotI restriction sites. Mel526 KIT KO and HepG2 WT were then transfected with KIT-3XFLAG encoding vector, and stable cells were prepared by G418 selection and sorting for KIT positive cells.

**Immunofluorescence**. Mel526 cells were plated into an eight-well chamber ($2 \times 10^4$ cells per well). On the following day the medium was replaced with a fresh medium containing the relevant chemical compounds and then incubated for 12 h. Treated cells were washed three times with cold PBS and fixated by adding 400 µl of 4% paraformaldehyde (diluted in PBS) for 10 min at room temperature. After washing with PBS, cells were permeabilized with 0.25% Triton X-100 for 10 min followed by BSA (2% in PBS) blocking for 30 min. Mouse anti-FLAG and goat anticalnexin primary antibodies were added (1:250) and incubated overnight at 4 °C. Cells were washed and goat anti-mouse-Alexa Fluor488 (ThermoFisher #A-11001) and donkey anti-goat cy3 (abcam #ab6949) were added (1:500) for 1 h in room temperature. Cells then were washed three times. PBS and 200 µl of DAPI mounting medium was added to each well (VECTASHIELD®, Vector Laboratories). Images were recorded by Olympus FV10i confocal microscope. All images were subsequently processed by Olympus fluoview fv1000 software.

**Flow cytometry**. Cells were harvested, centrifuged, washed twice in PBS, and resuspended in 100 µl of PBS. Fluorophore-conjugated antibody was added according to the manufacturer's instructions, followed by filtration through a 100 µM strainer directly to FACS tubes. Analysis was performed using Cytoflex FACS and CytExpert software for data processing. Conjugated antibodies used: APC-anti-human CD117 (KIT) (biogems #19211-80); APC-anti-human-c-MET (Sino Biological #10692-R271-A); Alexa 488-anti-human EGFR (BioLegend #352907), mouse anti HLA class I antibody (W6/32, abcam #ab22432).

**Protein stability assay**. Protein stability experiments were performed using CHX chase. Cells were harvested in 15 ml tubes, washed twice with cold PBS, and resuspended again in PBS. CHX (50 µg/ml) was added to each tube and immediately incubated in a 37 °C water bath during the entire experiment. Cells were homogenously re-suspended by pipetting at each time point, and the same volume was taken. Cells were centrifuged, and the cell pellets were kept at −80 °C until the end of the chase. Samples were processed for Western blotting as detailed above.

**Secreted proteins analysis**. A total of $1 \times 10^6$ HepG2 cells were plated on 6 cm plates for 8 h. Medium was then aspirated, and the cells were washed three times with PBS before the addition of serum-free DMEM with the indicated treatments. After 12 h, the supernatants were collected, and TCA-DOC protein precipitation was performed.

TCA-DOC protein precipitation: 1% (v/v) of a 2% (g/v) sodium deoxycholate solution was added to the supernatants, gently vortexed, and left on ice for 30 min.

Trichloroacetic acid was then added to the supernatants (1:10 v/v of 100% g/v TCA—final concentration is 10%) and kept on ice for 60 min after vortex. Proteins were precipitated by centrifugation at $15,000 \times g$ CFU for 15 min at 4 °C. Supernatants were aspirated, and the produced protein pellets were washed three times with ice-cold acetone. The pellets were air-dried in chemical hood and dissolved with 200 µl of boiling sample buffer. The same volumes (20 µl) were loaded into 12% SDS-PAGE gel for western blot analysis.

**Cell surface protein isolation.** Plasma membrane proteins were first biotinylated using a Pierce Cell Surface Protein Isolation KIT (ThermoFisher #89881). Following the biotin labeling, cells were resuspended in hypotonic buffer (0.2 mM EDTA, 1 mM $NaHCO_3$) at the density of $10^8$ cells/ml and left for 30 min at 4 °C to swell and burst. Cells were then further disrupted with 50 strokes of a B pestle in a dounce homogenizer, and the nuclei and remaining intact cells were spun out at $800 \times g$ for 10 min. The supernatant was collected and spun at $100,000 \times g$ for 1 h, yielding a cytosolic supernatant and crude total membrane pellet. The biotinylated protein was then selectively isolated from plasma membrane crude using a streptavidin-bound bead according to the kit instructions. The isolation process was validated by immunoblotting the total lysate vs. the isolated proteins against c-MET and HLA-A2 proteins.

**Quantitative proteomics of surface proteins.** Isolated cell surface proteins were isolated in triplicate per condition as described above and separated by SDS-PAGE. All six gel lanes were cut into three equal, corresponding pieces per lane and subjected to a standard tryptic in-gel digest[41]. Briefly, all gel pieces were destained using methanol/$H_2O$ (50/50), washed with 50 mM ammoniumbicarbonate (ABC) for 10 min, and then subjected to gel shrinking using acetonitrile (washing and shrinking was repeated once). After drying of the gel pieces in a speed vac, the proteins were reduced using 10 mM DTT in ABC for 45 min at 56 °C and then alkylated using 55 mM iodoacetamide in ABC (RT, in the dark, 30 min). The samples were washed once again with 50 mM ABC, followed by acetonitrile treatment and dried in a speed vac. The proteins were digested overnight with 50 µl (12.5 ng/µl) of mass spectrometry grade trypsin (Thermo Fisher Scientific). Subsequently, the resulting peptides were extracted from the gel pieces using 50% acetonitrile in 0.1% trifluoroacetic acid and desalted using homemade C18-Tips. Lyophilized peptides were resuspended in 5% acetonitrile/3% formic acid and subjected to nano LC-MS/MS analysis using a nano Ultimate 3000 liquid chromatography system and a Q Exactive plus mass spectrometer (both Thermo Fisher Scientific). Trapping of the peptides was performed on a precolumn (Acclaim PepMap100, C18, 5 µm, 100 Å, 300 µm i.d. × 5 mm, Thermo Fisher Scientific) for 10 min with buffer A (0.1% formic acid). The tryptic peptides were separated on an Easyspray C18 analytical column (2 µm particle size, 75 µm inner diameter, 25 cm length, 40 °C column oven temperature, 2 kV spray voltage; Thermo Scientific, Bremen, Germany) coupled to an Easyspray source (Thermo Scientific, Bremen, Germany) using a 140 min gradient: 0–10 min: 5% buffer B (80%ACN/0.1% FA), 10–104 min: 5–35% B, 104–114 min: 35–45% B, 114–114.1 min: 45–95% B, 114.1–119 min: 95% B, 119–120 min: 99–5% B, 120–140 min: 5% B.

The mass spectrometer was operated in data-dependent mode. Settings: resolution of 70,000, AGC of 3E6 ions, scan range 300–1600 $m/z$. dd-MS[2] settings: resolution of 17,500, AGC target 2e5, top 20 precursor fragmentation, collision energy: 27, dynamic exclusion 30 s.

**Data analysis and statistics of the comparative proteomic data.** The raw data were analyzed by using MaxQuant v1.6.3.3[42] and searched against the human Uniprot database version 01/2019 (only canonical and reviewed entries) using the Andromeda search engine with default mass tolerance settings[43]. Quantification was performed using the label-free quantification option in MaxQuant (LFQ) with default settings Trypsin (without proline limitation) was set as the sole protease with two allowed missed cleavages. Fixed modification: Carbamidomethylation; variable modifications: Methionine oxidation, Lysine modification (plus cleaved Sulfo-NHS-SS-Biotin): +87.998 (H(4) O C(3) S) and N-terminal protein acetylation. The false discovery rates were set to 0.01 for peptides, proteins, and modification sites; minimum peptide score for modified peptides: 40; minimum peptide length: seven amino acids. The resulting data were filtered for "potential contaminants," reverse and "only identified by site" entries. A minimum of two peptides (all) and one peptide (unique) was required. A protein needed to be quantified in all three replicates of at least one condition to be considered for further analysis. The median of all three LFQ intensities was calculated, and a minimum value of $5 \times 10^6$ of the LFQ intensity median in a least one of the two conditions (control "ctrl" or treatment "treat") was required for downstream data analysis. We used the Perseus software suite for statistical and bioinformatic analyses[44]. An FDR of <0.05 was employed to identify significantly up- and down-regulated proteins. For functional enrichment analysis, the DAVID webserver[45] was used using the human database as background.

**In vivo growth of HepG2 xenografts.** HepG2 cells ($2 \times 10^6$ cells in 100 µl PBS) were subcutaneously injected into NOD/SCID mice flanks (males at the ages of 2–3 months). On day 3, prior to the appearance of a palpable tumor, mice were divided according to the different treatments. Drug formulation and dosage

regime: All drugs were first dissolved in DMSO. ISRIB was dissolved at a concentration of 5 mg/ml, nelfinavir 100 mg/ml, and lopinavir 34 mg/ml. Prior to injection Tween 80 was added (1 µl per 5 µl of DMSO), and 1:10 dilution was performed with injectable saline. Drugs dosage was as follows: ISRIB 2.5 mg/kg, nelfinavir 50 mg/kg, and lopinavir 17 mg/kg. After 14 days of treatment mice were sacrificed.

**Ethics oversigh.** All mouse experiments were carried out under IACUC approved protocol MD-18-15472-4. HU is AAALAC approved.

**Plasma samples preparation and biochemical determination of AST/ALT levels.** Blood samples were first collected from mouse facial veins. Samples were then left undisturbed for 1 h at room temperature followed by centrifugation at $4000 \times g$ CFU for 10 min. The supernatants were then collected into new tubes. Prior to the biochemical quantification of AST and ALT levels, plasma samples where diluted 1:4 with saline and analyzed by Cobas C111.

**IHC and H&E staining.** Tissues were first fixed with 4% paraformaldehyde solution for 24 h, and then preserved in 70% ethanol solution until use. The preserved tissues were embedded into paraffin blocks, and 3 µm sections were made by microtome, and then placed on glass slides followed by deparaffinization and hydration processes. Heat-mediated antigen was retrieved with 10 mM citrate buffer pH 6.0 (Thermo Scientific, IL, USA #005000). Endogenous peroxide was inhibited by incubating the slides with a freshly prepared 3% $H_2O_2$ solution in methanol. Sections where blocked with 2.5% horse serum (VE-S-2000, Vector Laboratories #S-2000) for 1 h to diminish unspecific antigens bindings and to lower the staining background. Following the blocking process, slides were incubated with primary antibody in a dilution of 1:400 overnight at 4 °C. Antibodies used are: mouse antiphospho-Tyr (pY99) antibody (Santa Cruz Biotechnology, Inc. #sc-7020) and rabbit anti α1-antitrypsin (abcam #ab129354). On the following day, MACH3 HRP-polymer system (Biocare Medical, CA, USA) was used to detect and amplify the primary antibody signal. Visible staining was obtained by ImmPACT DAB (Vector, CA, USA #SK-4105). Counterstaining of nuclei was performed by hematoxylin staining (Vecmount, Vector laboratories, #H-5000). Tissue images were taken from random ×20 fields per each tissue using light microscope Zeiss AxioCam ICc5 color camera mounted on a Zeiss Axio Scope.

**Reporting summary.** Further information on research design is available in the Nature Research Reporting Summary linked to this article.

## Data availability
The mass spectrometry proteomics data have been deposited in the ProteomeXchange.
Consortium via the PRIDE partner repository with the data set identifier PXD014709[46]. All uncropped blots and original flow cytometry gating are provided as a Source Data File 1. Data used for Fig. 2d are provided as Source data files 1, 2. Further data are available on request from the authors.

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

## Acknowledgements

B.T. is the incumbent of the David Eisenberg Chair in Pharmacy. Research was funded by grants from the David R. Bloom Center for Pharmacy, the Dr. Adolph and Klara Brettler Center for Research in Pharmacology, Israel Cancer Association Grant No. 20200018 (B.T.), ISF Grants Nos. 696/2014 (B.T.), 1537/18, 1765/13 (D.R.), Treatment H2020-MSCA-ITN-721236 (B.T. and P.D.), START Program of the Medical Faculty of the RWTH Aachen University (T.W., 691517), German Israeli Fund (Grant no. I-1471-414.13/2018) to B.T. and M.H. M.M. is the recipient of the Zvi Yanai Doctoral Fellowship of the Israeli Ministry of Science and Technology.

## Author contributions

M.M., M.H., and B.T. designed the experiments. M.M., S.B., A.O., O.D., P.D., A.R., G.M., R.K., T.W. performed experiments. C.P. performed the MS analyses. R.F. and D.R. analyzed the MS data. B.T. wrote the manuscript.

## Competing interests

The authors declare no competing interests.
