## [Peer Review File · Nature Communications]

Reviewers' comments:

Reviewer #1 (Remarks to the Author):

This work is focused in investigating the impact of the pharmacological inhibition of PERK/IRS in the context to ER stress to alter the normal maturation and trafficking of several RTKs as a possible new therapy in the cancer treatments. Mohameed and colleagues performed a well-designed study. Data regarding alterations in the selective ER retention is very impressive and the concept of -selective ER retention- is novel and clever. However, the paper abuse of correlations without providing a clear mechanism and a lack of a well extended discussion about it. Also some of the conclusions are overestimated as the regulation of PERK p-Tyr in the cells without specific controls as downstream effectors of MET, EGFR OR KIT as an example. For prove of concept it will be important to use some cells lines addicted to some RTKs overexpression/mutations that confer its hiperactivation like HER2 cells lines (Breast cancer) or lung cancer cells lines with EGFR overexpression. Until now the only drug that inhibits kinases used in HCC is sorafenib a pankinase inhibitor and the "oncogenic addiction" to some mutated RTKs it's not clear like others cancers. Better discussion of the available literature is necessary. Overall, the experimental work from the Tirosh lab is well performed, the research group used different biochemical approaches. If the authors are suitable to answer the following queries the study would be recommended to be published in the journal.

-FIG 1A please use a positive control of p-Tyr activation and a inhibitory drug to see the dynamics of the process. Test a direct target that is phosphorylated by RTK as control

-FIG 1C tm treatments affect the total levels of PERK lane 3..lane 7 the treatments with GSK reduce a lot the expression of KIT. Please explain.

Fig1E lane 1 and 3 the signal are very similar and it is not clear a real effect...lane 2 and 8 GKS decrease the quantity of KIT independent of the PERK expression?

Figure S1B please add a quantification with proper statistical analysis.

Is it possible to detect a similar effect in the retention and maturation of RTKs with ATF6 deletion like IRE1 in your model?

Figure 2c the number of counts is very dissimilar.

3c d the cell viability experiments are not well performed, the number of cells are dissimilar even in the same treatments HEPG2 DMSO Right and left please repeat the experiments.

Also the arguments of the differences in the retention of RTKs by Tg in mouse and human cells is very weak please explain better these results.

Reviewer #2 (Remarks to the Author):

This manuscript showed that PERK inhibition together with ER stress-inducers caused selective ER retention of RTKs including KIT, c-MET and EGFR. In addition, the authors applied this effect to anti-cancer therapy, suggesting that findings in this study may have clinical implications as well as basic significance. This manuscript contains several novel findings and the experimental design is appropriate to test their hypothesis. I have some comments as follows.

1. The authors definitively proved that PERK inhibition and ER stress-inducers cooperatively caused selective ER retention of membrane-associated proteins. They also showed that ATF4, a downstream target of PERK, did not play a role in this process. However, it remained still unclear how PERK is implicated in the selective ER retention. PERK should have several ATF4-independent functions. For example, PERK was reported to be involved in ER-plasma membrane contact and ER-mitochondrial contact in an ATF4 signaling-independent manner (van Vliet, et al. Mol Cell 2017, Verfaillie, et al. Cell Death and Diff 2012). I think that to clarify this issue, at least in part, is important for this study.
2. Because a lot of proteins were downregulated by Tg/GSK414 treatment (Figure 2E), a therapy targeting selective ER retention may affect a wide range of cellular functions. However, the authors showed that this treatment had minimal toxicity in normal cells. Is there any explanation in this phenomenon?
3. As shown in Figure 2E, proteins involved in cell-cell adhesion which are the most popular membrane-associated proteins were decreased by Tg/GSK414 treatment. In fact, cells shown in Figure S1B and Figure S8 seemed to lose cell-cell attachment and alter cellular morphology. If this treatment strategy affects cell-cell attachment, adverse effects on the epithelial barrier such as gastrointestinal tract may be concerned in vivo. Did the authors have any data about the cell adhesion molecules after Tg/GSK414 treatment and the effect of nelfinavir/ISRIB treatment on the gastrointestinal tract?
4. In this study, nelfinavir/ISRIB treatment did not cause liver toxicity. However, as the authors mentioned in this paper, dysfunction of PERK is known to be associated with liver injury. Additionally, most human HCCs arise in the context of chronic liver injury and even cirrhosis. Therefore, liver-related adverse effect should be still noted.

Dear reviewers,

We would like to thank you for the insightful comments. Importantly, each of you highlighted different aspects, thus by addressing all comments the paper is now significantly improved. The major changes in the revised manuscript are:

- A new figure was added to the manuscript body and a new figure (S1) was added to the supplemental material.
- Figure 1C and 1D were revised.
- A new panel was added to Figure 1 (panel E), which shows the effect of the Tg/GSK and Tg/ISRIB on the phosphorylation status of KIT relative to dasatinib, as requested by the reviewers.
- Figure 1F of the original version was moved to the new figure S1 and a new panel that analyzes the role for ATF6 in sERr was added.
- A new panel was added to Figure 3, which shows the rescue effect of translation attenuation to sERr. This is an important mechanistic addition to the paper as was requested by the reviewers.
- Figure 4 was broken into two figures, 4 and 5. Two new experiments with RTK-addicted cells were added to figure 4. The comparison between HepG2 and hTERT is now shown in a separate figure, figure 5. The experiment was repeated and the growth curves were revised.
- We quantified the co-localization analyses by Pearson's coefficient and added it to the supplemental material (now Figure S2).

Overall, these additions address all the requested experiments by the reviewers.

Enclosed is our response in a point-by-point manner.

Reviewer #1:

This work is focused in investigating the impact of the pharmacological inhibition of PERK/IRS in the context to ER stress to alter the normal maturation and trafficking of several RTKs as a possible new therapy in the cancer treatments. Mohamed and colleagues performed a well-designed study. Data regarding alterations in the selective ER retention is very impressive and the concept of -selective ER retention- is novel and clever. However, the paper abuse of correlations without providing a clear mechanism and a lack of a well extended discussion about it. Also some of the conclusions are overestimated as the regulation of PERK p-Tyr in the cells without specific controls as downstream effectors of MET, EGFR OR KIT as an example.

We think that the word "abuse" is inappropriate in this context. We have tried hard not to over interpret our data. We do agree with the reviewer that a comparison to clinically used kinase inhibitors is important to show and also to monitor the phosphorylation status of some RTKs directly. This has been done according to the specific comments below and added to the revised manuscript.

For prove of concept it will be important to use some cells lines addicted to some RTKs overexpression/mutations that confer its hyperactivation like HER2 cells lines (Breast cancer) or lung cancer cells lines with EGFR overexpression. Until now the only drug that inhibits kinases used in HCC is sorafenib a pankinase inhibitor and the "oncogenic addiction" to some mutated RTKs it's not clear like others cancers.

Per this specific comment we have done several experiments. First we analyzed the phosphorylation of KIT and total phospho-tyrosine in the MEL526 cells in response to Tg/GSK414 and Tg/ISRIB combinations, and dasatinib as a positive control. Second, we analyzed the phosphorylation of MET and EGFR and survival of EGFR-addicted lung cancer cell lines with nelfinavir and lopinavir in combination with ISRIB. In this case the EGFR inhibitor osimertinib was used as a positive control. Finally, we used the KIT-addicted HMC1.1 cells with dasatinib as the positive control. These experiments undoubtedly improved the manuscript, and we thank the reviewer for suggesting them.

Better discussion of the available literature is necessary. Overall, the experimental work from the Tirosh lab is well performed, the research group used different biochemical approaches. If the authors are suitable to answer the following queries the study would be recommended to be published in the journal.

We appreciate the overall positive assessment of our study. We improved the discussion and hope that the revision will satisfy the reviewer.

-FIG 1A please use a positive control of p-Tyr activation and an inhibitory drug to see the dynamics of the process. Test a direct target that is phosphorylated by RTK as control

The focus of figure 1 is on KIT. Based on the reviewer's comment we added an immunoblot analysis of KIT/P-KIT and total P-Tyr also with dasatinib (BMS-354825), a multi-targeted kinase inhibitor that inhibits wt and oncogenic KIT mutants (PMID 16397263). This is labeled as **Fig. 1E**. The addition of dasatinib completely blocked the phosphorylation of KIT when triggered with SCF and reduced the total cellular P-Tyr levels similar to the Tg/GSK414 treatment. We think that this is an important control to include and thank the reviewer for this suggestion.

-FIG 1C tm treatments affect the total levels of PERK lane 3..lane 7 the treatments with GSK reduce a lot the expression of KIT. Please explain.

Indeed, KIT levels are reduced by GSK414. This is due to acceleration of its lysosomal degradation as we recently published (Mahameed M *et al.*, Cell Death Dis. 2019 Apr 1;10(4):300). This is an off-target pharmacological activity of GSK414. We refer to this paper in the text, but we need to better emphasize this off-target effect of GSK414. This has not been observed with other PERK inhibitors such as AMG PERK 44. For the experiment with HMC1.1 cells we used this inhibitor to avoid the off-target toxicity of GSK414 as will be explained below.

Fig1E lane 1 and 3 the signal are very similar and it is not clear a real effect...lane 2 and 8 GSK decrease the quantity of KIT independent of the PERK expression?

Indeed. Tg alone has a minimal effect on the KIT level. GSK414 reduces KIT levels independently of its effect on PERK as we have published.

Figure S1B please add a quantification with proper statistical analysis.

Thank you for the comment. We have analyzed 30 different cells for each treatment and added the quantification of co-localization evaluated by Pearson's correlation coefficient. Data clearly show the ER localization of KIT with Tg/GSK414 and Tg/ISRIB treatments.

Is it possible to detect a similar effect in the retention and maturation of RTKs with ATF6 deletion like IRE1 in your model?

The mechanistic analysis added to the revised paper shows that the retention in the ER is most likely due to the translation regulation by PERK (revised Fig. 3). Thus, the likelihood that ATF6 has a role in the retention is low. Due to two active isoforms one needs to generate double KO cells to generate ATF6 null cells. However, per the comment of the reviewer we blocked ATF6 activation pharmacologically by the protease inhibitor, pefabloc, that inhibits regulated intramembrane proteolysis (RIP) of ATF6 (PMID 12782636). The figure added to the revised manuscript shows a lack of effect for pefabloc in the ER retention of KIT. We thank the reviewer for his comment.

Figure 2c the number of counts is very dissimilar.

We corrected the number of events in the flow cytometry acquisition. We revised figure 2c accordingly. Thank you for the comment.

3c d the cell viability experiments are not well performed, the number of cells are dissimilar even in the same treatments HEPG2 DMSO Right and left please repeat the experiments.

The experiment was repeated. Figure 3C and D were revised. Thank you for the comment.

Also the arguments of the differences in the retention of RTKs by Tg in mouse and human cells is very weak please explain better these results.

This was an unexpected observation. Apparently Tg alone in these cells was sufficient to retain MET in the ER, thus PERK inhibition should not have an additional effect. Although we do not understand the molecular mechanisms that underlies this difference, we addressed it in the discussion. One possible difference can be ERp44, itself. In contrast to human cells in which ERp44 is O-glycosylated, in mouse cells ERp44 is not. To explore whether this may be the reason, we need to reconstitute the mouse cells with the human ERp44. At this point in time, we just wanted to report this fundamental difference between the mouse cells we analyzed and human HCC cells. These data can be removed from the manuscript if deemed unnecessary by the reviewer.

Reviewer #2:

This manuscript showed that PERK inhibition together with ER stress-inducers caused selective ER retention of RTKs including KIT, c-MET and EGFR. In addition, the authors applied this effect to anti-cancer therapy, suggesting that findings in this study may have clinical implications as well as basic significance. This manuscript contains several novel findings and the experimental design is appropriate to test their hypothesis. I have some comments as follows.

1. The authors definitively proved that PERK inhibition and ER stress-inducers cooperatively caused selective ER retention of membrane-associated proteins. They also showed that ATF4, a downstream target of PERK, did not play a role in this process. However, it remained still unclear how PERK is implicated in the selective ER retention. PERK should have several ATF4-independent functions. For example, PERK was reported to be involved in ER-plasma membrane contact and ER-mitochondrial contact in an ATF4 signaling-independent manner (van Vliet, et al. Mol Cell 2017, Verfaillie, et al. Cell Death and Diff 2012). I think that to clarify this issue, at least in part, is important for this study.

Thank you for the comment. We titrated CHX to artificially fine tune protein synthesis. Data were incorporated into the revised manuscript as Figure 3. As can be seen, attenuation of protein synthesis restored the maturation of MET and resulted in a slight elevation (rather than reduction) of surface MET and EGFR levels. This experiment provides strong indications that the underlying reason for the retention is primarily the result of a misbalanced in protein synthesis as conferred by the PERK inhibitor or ISRIB. We greatly appreciated the reviewer's comment. This is in our opinion a significant addition to the paper.

2. Because a lot of proteins were downregulated by Tg/GSK414 treatment (Figure 2E), a therapy targeting selective ER retention may affect a wide range of cellular functions. However, the authors showed that this treatment had minimal toxicity in normal cells. Is there any explanation in this phenomenon?

It should be emphasized that the proteomic analysis was performed on Tg vs Tg/GSK414 treatments while *in vivo* we used nelfinavir/ISRIB. The effects of nelfinavir/ISRIB are milder than Tg/GSK414. There is no doubt that toxicity will also be developed in normal cells, however our data suggest that there might be a therapeutic window for selectivity. We emphasized this difference in the discussion and results sections.

3. As shown in Figure 2E, proteins involved in cell-cell adhesion which are the most popular membrane-associated proteins were decreased by Tg/GSK414 treatment. In fact, cells shown in Figure S1B and Figure S8 seemed to lose cell-cell attachment and alter cellular morphology. If this treatment strategy affects cell-cell attachment, adverse effects on the epithelial barrier such as gastrointestinal tract may be concerned *in vivo*. Did the authors have any data about the cell adhesion molecules after Tg/GSK414 treatment and the effect of nelfinavir/ISRIB treatment on the gastrointestinal tract?

This comment is a continuation of the previous one. Again, Tg/GSK414 treatment is toxic, and all cell lines that we tested succumb to cell death with this treatment after a certain amount of time. This was the reason why we had to look for less toxic alternatives. Per the comment of the reviewer, we subjected NOD/SCID mice to 14 days of treatment with nelfinavir, ISRIB or their combination and analyzed the small intestine by histology. The analysis that we made on three mice from each group did not show any sign of alteration to normal intestine histology. We added the H&E images of the combined nelfinavir/ISRIB to this letter, but we do not think they should be incorporated into the body of the manuscript. We raised this issue in the revised discussion. We thank the reviewer for this comment.

4. In this study, nelfinavir/ISRIB treatment did not cause liver toxicity. However, as the authors mentioned in this paper, dysfunction of PERK is known to be associated with liver injury. Additionally, most human HCCs arise in the context of chronic liver injury and even cirrhosis. Therefore, liver-related adverse effect should be still noted.

To our knowledge there are no indications of liver toxicity when PERK is compromised in mice. In fact, the opposite was documented. Liver function improved when PERK was inhibited (PMID 26435271, PMID 28869608). We should like again to emphasize that we have not used the PERK inhibitor *in vivo* for issues of potential toxicity to the pancreas. We added the reviewer's reservation to the discussion and indicated that the combination might result in liver toxicity, however we still do not have any indication that this is indeed an issue.

We do not claim in the paper that induction of sERr is the holy grail of cancer treatment. Certainly, there are clinical limitations to this pharmacological approach. However, this is a new biological phenomenon that can be induced by drugs and has a potential to be developed into therapy, probably in conjunction with TKIs.

REVIEWERS' COMMENTS:

Reviewer #1 (Remarks to the Author):

After the revision the authors addresses all the relevant points requested to improve this manuscript. Also all the new work made by Tirosh's group further increased the quality and clarity of the study. I recommended this work for publication however it's necessary to addressed the following minor points:

First, I recommended excluding the mouse cell line data, because it doesn't add more clarity to the new findings and is confusing.

Second, I have some queries as:

Lack of statistics (figure 5b)

Is it possible to show the figures 4D and 4F (Cell death) more friendly? A better way to see the differences and the significance between the groups?

It seems there is more protein the in the group of Nelfinavir+ISRIB when the loading control is compared (Figure 6B)

Could you improve the EGFR blots (Figure 6B) to detect the immature form?

Some typo inconsistencies as SERR (Line 569)

Reviewer #2 (Remarks to the Author):

The authors have mostly addressed previous my concerns and the manuscript is improved.

Dear reviewer #1,

Thank you for your comments. Enclosed is our response. In essence we accepted all comments and revised our manuscript accordingly.

REVIEWERS' COMMENTS:

Reviewer #1 (Remarks to the Author):

After the revision the authors addresses all the relevant points requested to improve this manuscript. Also all the new work made by Tirosh's group further increased the quality and clarity of the study. I recommended this work for publication however it's necessary to address the following minor points:

First, I recommended excluding the mouse cell line data, because it doesn't add more clarity to the new findings and is confusing.

The data was removed from the manuscript and the relevant text in the discussion was removed as well.

Second, I have some queries as:

Lack of statistics (figure 5b).

Statistics was added.

Is it possible to show the figures 4D and 4F (Cell death) more friendly? A better way to see the differences and the significance between the groups?

A bar graph was added. Exact numbers appear on the dot plots.

It seems there is more protein in the group of Nelfinavir+ISRIB when the loading control is compared (Figure 6B). Could you improve the EGFR blots (Figure 6B) to detect the immature form?

The experiment was repeated. EGFR is not resolved well from the tumor extracts, perhaps owing to a background signal from the tumor mouse stroma. However, retention of c-MET is clearly observed. Western was replaced and the retention of c-MET was quantified.

Some typo inconsistencies as SERR (Line 569)

Corrected. Thanks.